# The Biology, Impact, and Management of *Xyleborus* Beetles: A Comprehensive Review

**DOI:** 10.3390/insects15090706

**Published:** 2024-09-17

**Authors:** Sared Helena Rodríguez-Becerra, Rafael Vázquez-Rivera, Karla Irazú Ventura-Hernández, Tushar Janardan Pawar, José Luis Olivares-Romero

**Affiliations:** 1Red de Estudios Moleculares Avanzados, Instituto de Ecología, A.C. Carretera Antigua a Coatepec 351, Xalapa 91073, Veracruz, Mexico; saredh_rodbe@hotmail.com (S.H.R.-B.); rafarivera2001@hotmail.com (R.V.-R.); irazuhdez1995@gmail.com (K.I.V.-H.); 2Facultad de Química Farmacéutica Biológica, Universidad Veracruzana, Circuito Gonzalo Aguirre Beltrán s/n, Zona Universitaria, Xalapa 91090, Veracruz, Mexico; 3Instituto de Química Aplicada, Universidad Veracruzana, Av. Luis Castelazo Ayala s/n, Col. Industrial-Animas, Xalapa 91190, Veracruz, Mexico

**Keywords:** *Xyleborus* beetles, ambrosia beetles, forest pests, agricultural impact, pest management, biological control, integrated pest management (IPM), molecular genetics, ecological impact, economic impact

## Abstract

**Simple Summary:**

*Xyleborus* beetles are tiny insects that impact forests and agriculture. These beetles burrow into trees and crops, causing damage that affects both the environment and the economy. This review aims to provide a clear understanding of the biology and behavior of *Xyleborus* beetles, including their life cycle, habitat, and feeding habits. We also discuss the various methods currently used to manage and control these pests, such as chemical treatments, biological controls, and integrated pest management strategies. By highlighting recent scientific advancements and identifying future research needs, this review offers insights for researchers, farmers, and policymakers. Understanding and effectively managing *Xyleborus* beetles is crucial for protecting our forests and agricultural resources, ultimately benefiting society by reducing economic losses and preserving ecological balance.

**Abstract:**

*Xyleborus* beetles, a diverse group of ambrosia beetles, present challenges to forestry and agriculture due to their damaging burrowing behavior and symbiotic relationships with fungi. This review synthesizes current knowledge on the biology, ecology, and management of *Xyleborus*. We explore the beetles’ life cycle, reproductive strategies, habitat preferences, and feeding habits, emphasizing their ecological and economic impacts. Control and management strategies, including preventive measures, chemical and biological control, and integrated pest management (IPM), are critically evaluated. Recent advances in molecular genetics and behavioral studies offer insights into genetic diversity, population structure, and host selection mechanisms. Despite progress, managing *Xyleborus* effectively remains challenging. This review identifies future research needs and highlights innovative control methods, such as biopesticides and pheromone-based trapping systems.

## 1. Introduction

The genus *Xyleborus* is a group of ambrosia beetles belonging to the subfamily *Scolytinae* within the family *Curculionidae*. These beetles are known for their unique symbiotic relationships with fungi, which they cultivate within the galleries they bore into wood. This mutualistic association is a defining characteristic of ambrosia beetles, distinguishing them from other bark beetles that primarily feed on the phloem of trees. The name “ambrosia” refers to the fungal “gardens” these beetles create and tend, which serve as their primary food source. The denomination of “bark beetle” originates from the fact that many species feed on the inner bark (phloem) layer of trees. However, this subfamily also includes numerous species with different lifestyles, such as those that bore into wood, feed on fruits and seeds, or tunnel into herbaceous plants [1]. This intricate interaction has ecological and economic implications, making *Xyleborus* a key subject of study in both forestry and agriculture [2,3]. *Xyleborus* is a genus of beetles classified under the domain Eukaryota and belongs to the kingdom Animalia. It is part of the phylum Arthropoda, which encloses animals with jointed limbs and exoskeletons. Within the class, Insecta, *Xyleborus* is categorized under the order Coleoptera, commonly known as beetles. This genus is specifically part of the family *Curculionidae*, which includes weevils and belongs to the subfamily *Scolytinae*, known for bark beetles. The tribe Xyleborini contains the genus *Xyleborus*, which is recognized for its wood-boring behavior.

The study of *Xyleborus* dates to the early 19th century when entomologists first began to document the diversity and behavior of bark and ambrosia beetles [4]. Over the decades, extensive research has been conducted to understand their taxonomy, life cycles, and the intricate relationships they maintain with their fungal symbionts. Early work focused primarily on identifying species and describing their morphological traits, but advancements in molecular biology and genetics have since revolutionized our understanding of these beetles. The ability to sequence DNA and analyze genetic material has provided deeper insights into the evolutionary relationships among *Xyleborus* species and their symbiotic fungi [5,6,7]. 

The study of *Xyleborus* beetles is crucial due to their ecological and economic impacts. These beetles are notorious for their ability to infest a wide range of host plants, including economically relevant timber and fruit trees [2,3,5]. Their burrowing activity can cause damage, leading to reduced wood quality, decreased crop yields, and even tree mortality. This not only results in economic losses for forestry and agricultural industries but also affects ecosystem stability and biodiversity [8]. The symbiotic relationship between *Xyleborus* beetles and their fungal partners adds another layer of complexity. The fungi cultivated by these beetles can be pathogenic to plants, exacerbating the damage caused by the beetles themselves [7]. Understanding the dynamics of this relationship is crucial for developing effective control measures.

Despite their destructive potential, *Xyleborus* beetles also play important roles in their natural ecosystems. As decomposers, they contribute to the breakdown of dead and dying wood, facilitating nutrient cycling and promoting forest health. Their galleries provide habitats for other organisms, including fungi, bacteria, and other invertebrates. This intricate web of interactions underscores the importance of a balanced perspective when studying *Xyleborus* beetles. Although they can be pests in managed environments, they also fulfill essential ecological functions in natural settings [9,10,11,12].

The economic impact of *Xyleborus* beetles is severe, particularly in regions where forestry and agriculture are major economic drivers. In the forestry sector, infestations can lead to losses in timber quality and quantity [13,14]. Beetle-infested wood often suffers from staining and structural damage, reducing its market value. In extreme cases, large-scale infestations can necessitate costly management interventions, including tree removal and replacement. In agriculture, *Xyleborus* beetles pose a threat to fruit trees and other crops [11]. For example, infestations in avocado and mango plantations can lead to reduced yields and increased production costs [15,16]. The global trade in these commodities means that infestations can have far-reaching economic consequences, affecting not only local producers but also international markets.

Effective management of *Xyleborus* beetles requires a multifaceted approach. Preventive measures, such as monitoring and early detection, are critical for managing infestations before they become unmanageable [17,18,19]. Chemical treatments, while effective in some cases, must be used judiciously to minimize environmental impact and the development of resistance [20,21,22,23,24,25]. Biological control methods, including the use of natural predators and entomopathogenic fungi, offer promising alternatives to chemical treatments [26]. Integrated pest management (IPM) strategies, which combine multiple control methods, are increasingly being recognized as the most effective approach for sustainable management [27].

Recent advances in molecular genetics have opened new avenues for research on *Xyleborus* beetles. Techniques such as DNA barcoding and genome sequencing are providing detailed insights into the genetic diversity and population structure of these beetles [5,28,29]. Behavioral studies are shedding light on their mating and dispersal strategies, while innovative control methods, such as pheromone-based trapping systems, are being developed [15,19,30,31,32]. Despite these advances, challenges remain. The complexity of *Xyleborus* biology and their interactions with symbiotic fungi means that there is still much to learn. Future research needs to focus on understanding these interactions in greater detail, as well as developing more sustainable and ecologically sound management practices.

This review aims to provide a comprehensive synthesis of the current knowledge on *Xyleborus* beetles, with a focus on their biology, ecological roles, and the challenges they pose to forestry and agriculture. The objectives are to describe the biology and ecology of *Xyleborus* beetles, evaluate their economic and ecological impact, critically assess current control and management strategies, highlight recent advances in *Xyleborus* research, and identify future research needs. By synthesizing current knowledge and identifying future research needs, this review aims to provide a comprehensive resource for researchers, practitioners, and policymakers. Effective management of *Xyleborus* beetles will not only mitigate economic losses but also contribute to the health and stability of forest ecosystems.

## 2. Biology and Ecology of *Xyleborus*

### 2.1. Life Cycle and Reproduction

The life cycle and reproduction of *Xyleborus* beetles involve the construction of galleries, fungal inoculation, larval development, and adult dispersal (Figure 1). Their reproductive strategy is characterized by inbreeding and a close symbiotic relationship with fungi, which are critical for their survival and success [33]. 

#### 2.1.1. Host Tree Selection

Most ambrosia beetles colonize stressed, weakened, dying, and dead trees [17], using olfactory signals (mainly ethanol) [34] and visual cues in their search for suitable hosts for colonization and reproduction. These beetles are attracted to trees with larger stem diameters [35]. It is still not entirely clear why some ambrosia beetles, such as *X. glabratus*, infest healthy trees. It has been observed that this beetle is attracted to volatile compounds produced by Lauraceae, such as α-copaene and calamenene, and that it attacks smaller stems [19,36,37].

In an experiment conducted by Martini et al., it was observed how the volatile compounds generated by *Persea palustris*, inoculated with *R. lauricola*, play an important role in the attraction and repulsion of *X. glabratus* to the plant. In three days after infection, *X. glabratus* is repelled by infected leaf odors due to an increase in methyl salicylate, a known behavioral repellent. However, at 10 and 20 days, the beetles are more attracted to the infected leaf odors compared to non-infected plants. This increased attraction is associated with a rise in sesquiterpenes and aldehydes in the leaf volatiles [38]. Additionally, compounds such as eucalyptol, cubeb, and α-copaene, which are known attractants, are present in higher amounts in infected leaves during these later stages [39,40]. These changes in volatile profiles are linked to the activation of the salicylic acid pathway by the fungal infection, which initially increases methyl salicylate release and later alters other volatile emissions, thereby impacting *X. glabratus* behavior [38].

#### 2.1.2. Egg Laying and Gallery Construction

This life cycle of *Xyleborus* beetles begins when a female selects a suitable host tree and starts to bore into the wood to construct a gallery system. These galleries are intricately designed tunnels that serve as both a breeding site and a nursery for the fungal symbionts. The female beetle uses her mandibles to chew through the wood, creating a network of tunnels. She lays her eggs in these galleries, which are strategically placed to optimize the growth of both the beetle larvae and the fungal garden [3,14].

#### 2.1.3. Fungal Inoculation

A critical aspect of the gallery construction is the inoculation of the tunnels with fungal spores. Female *Xyleborus* beetles carry these spores in specialized structures called mycangia, which are located near their mandibles or in other parts of their bodies, depending on the species. As the female excavates the galleries, she releases these spores into the wood. The spores quickly germinate and proliferate within the moist, nutrient-rich environment of the galleries, forming a dense fungal mat [14,41].

Larvae grow by feeding on symbiotic fungi seeded by their mother, but during the pupal stage, the contents of their gut are emptied. Then, the naïve adults acquire their own symbionts through feeding [14,41].

Environmental factors such as climate type, host plant, microbes and their metabolism and arthropods have been observed to affect the fungal diversity within the mycangia of Xyleborus [30,42,43,44,45]. In general, it has been observed that *Raffaelea* species constitute the primary symbionts of these ambrosial beetles. However, species from the orders *Hypocreales* and *Saccharomycetales* have also been found, demonstrating a broad diversity within the mycangial communities [46]. A study conducted during the life cycle of *X. affinis* suggests that the yeasts and bacteria contained in the beetles’ microbiome are closely involved in the cultivation of filamentous fungi such as *Raffaelea* [44]. These studies indicate that further research is still poorly understood the biological mechanisms involved in these mutualistic relationships.

#### 2.1.4. Larval Development

The eggs laid by the female beetle hatch into larvae, which are entirely dependent on the fungi for nourishment. The larvae feed on the growing fungal mycelium, which provides them with all the necessary nutrients for their development. This relationship is highly specialized, as the fungi break down complex wood polymers into simpler compounds that are easier for the larvae to digest. The larvae undergo several molts as they grow, eventually reaching the pupal stage within the safety of the gallery system [14,41].

#### 2.1.5. Pupal Stage and Emergence

After completing their larval development, the beetle larvae enter the pupal stage. During this stage, they undergo metamorphosis, transforming into adult beetles. The pupal stage is a period of physiological and morphological changes, preparing the beetles for their life outside the galleries. Once metamorphosis is complete, the new adult beetles emerge from the pupal cases and continue to reside in the galleries until they are ready to disperse [2,3,14,41].

#### 2.1.6. Reproductive Strategy and Inbreeding

One of the most notable features of *Xyleborus* beetles’ reproductive strategy is their reliance on inbreeding. The sex ratio within the galleries is heavily skewed towards females, with males being produced in much smaller numbers. Males are typically haploid and smaller than females. This skewed sex ratio facilitates brother–sister mating within the confined space of the galleries. The inbreeding strategy ensures that the fungal symbionts are passed on to the next generation without the need for the beetles to find new fungal partners, thus maintaining the stability of the mutualistic relationship [2,41].

#### 2.1.7. Generational Overlap and Population Dynamics

*Xyleborus* beetles often exhibit overlapping generations within a single gallery system. This means that different life stages (eggs, larvae, pupae, and adults) can coexist within the same gallery. Most *Xyleborus* species like *X. dispar*, *X. affinis*, and *X. glabratus* reproduce by arrhenotokous parthenogenesis, where males are rare, and females can reproduce without fertilization. In this system, females produce unfertilized eggs that develop into males, while fertilized eggs develop into females. This allows them to maintain high populations even in the absence of males, contributing to their success as pests. The population dynamics of *Xyleborus* beetles are thus closely linked to the health and availability of their host trees, as well as the successful cultivation of their fungal symbionts [14,47,48,49].

#### 2.1.8. Ecological and Evolutionary Implications

The reproductive strategies and life cycle of *Xyleborus* beetles have ecological and evolutionary implications. The reliance on inbreeding and the tight association with fungal symbionts have led to highly specialized and co-evolved relationships. These beetles have evolved to exploit a niche that is relatively inaccessible to other insects, giving them a competitive advantage. However, this specialization also makes them vulnerable to disruptions in their environment, such as the loss of suitable host trees or changes in the composition of their fungal partners [2,43].

### 2.2. Habitat and Distribution

The habitat and distribution of *Xyleborus* beetles are shaped by a combination of environmental factors, host tree availability, reproduction, and their symbiotic relationships with fungi. Their ability to adapt to various environments and colonize a wide range of tree species has facilitated their global distribution and, in some cases, their status as invasive pests, especially those beetle species that are capable of colonizing live trees [2,50,51]. 

Table 1 provides a compilation of the global distribution of *Xyleborus* species that affect economically important plants. *Xyleborus* beetles are distributed globally, with species occurring in a wide range of climatic zones, from tropical rainforests to temperate woodlands. They are particularly diverse and abundant in tropical and subtropical regions, where the warm and humid conditions favor the growth of their fungal symbionts and the decomposition of wood [2,3,48,52].

The habitat preferences of *Xyleborus* beetles are primarily influenced by the availability of suitable host trees and the presence of their symbiotic fungi [53]. These beetles typically inhabit forested areas, including both natural forests and managed plantations. They prefer environments where dead or dying trees are abundant, as these provide the ideal conditions for gallery construction and fungal growth [7,14,54]. 

**Table 1 insects-15-00706-t001:** Distribution and fungal association of economically important Xyleborus.

*Xyleborus* Species	Fungus-Associated	Affected Plants	Distribution
*Xyleborus affinis*	*Fusarium oxysporum*	*Archidendron clypearia* (*Jack*) *Benth*. [55,56]	Global [25,26,33,57,58,59,60]
	* Ceratocystis fimbriata *	*Manguifera indica* [2,61]	
	* Raffaelea lauricola *	*Persea americana* *Persea borbonia* *Persea palustris* *Sassafras albidum* *Lindera benzoin* *Cinnamomum camphora*	
*Xyleborus dispar*	* Ophiostoma novo-ulmi *	* Dutch elm disease *	Northern Europe [62]
*Xyleborus glabratus*	* Raffaelea lauricola *	*Persea americana* *Persea borbonia* *Persea palustris* *Sassafras albidum* *Lindera benzoin* *Cinnamomum camphora*	Asia, North America, [9,36,63]
*Xyleborus* *bispinatus*	Unknown	*Ficus carica* L.	Italy [64]
	* Raffaelea lauricola *	*Persea americana*	Mexico [3,65,66]
			Florida, USA [8,37]
	Unknown	*Apate monachus*	Iberian Peninsula [67]
*Xyleborus* *volvulus*	* Raffaelea lauricola *	*Persea americana*	From United States to South America. It is also found in Africa and Asia [9,47,49,68]
*Xyleborus* *perforans*	*Fusarium parceramosum*	*Pinnus* spp.	America, Europe, and Australis [69]
	*Fusarium aff. solani*		
	*Ophiostoma ips*		
	*Raffaelea deltoideospora*		
	*Sporothrix pseudoabietina*		
*Xyleborus* *ferrogineus*	*Ceratocystis cacaofunesta*	*Theobroma cacao* L. [51]	Latin America and Africa [60,69]

#### Invasive Species and Range Expansion

Several *Xyleborus* species have become invasive outside their native ranges, causing ecological and economic damage. These invasions are often facilitated by human activities, such as the global trade of timber and ornamental plants, which can inadvertently transport beetles and their fungal symbionts to new areas [42]. Once established in a new environment, invasive *Xyleborus* species can spread rapidly, infesting local tree populations and outcompeting native bark and ambrosia beetles.

One notable example is *X. glabratus*, which was introduced to the southeastern United States and has since spread throughout the region, causing widespread mortality in redbay and avocado trees due to laurel wilt disease, a condition caused by its fungal symbiont *Raffaelea lauricola* [52]. Similarly, *X. similis* has been reported in various countries outside its native range, affecting a wide range of tree species and contributing to ecological disruptions [42].

Another example is *X. perforans*, which has as sociated fungi such as *Fusarium parceramosum*, *Fusarium aff. solani*, *Ophiostoma ips*, *Raffaelea deltoideospora*, and *Sporothrix pseudoabietina*. These fungi affect *Pinus* spp. trees, causing economic and ecological impact [52].

Another study evaluates the potential invasion of the exotic Asian ambrosia beetle, *Xyleborus glabratus*, in Mexico and the southern United States. Using ecological niche modeling and spatial multi-criteria evaluation, it identifies high-risk regions based on climate suitability, forest stress factors, and the presence of susceptible host species, primarily within the Lauraceae family. Interestingly, the results suggest that these beetles, which carry pathogenic fungi, pose a significant threat to both native and cultivated forest ecosystems, such as avocado crops [70].

### 2.3. Natural Enemies

The natural enemies of the *Xyleborus* beetles, which are part of the ambrosia beetle group, include a variety of predators, parasitoids, and pathogens. These natural enemies play a crucial role in controlling beetle populations in their natural habitats. Below is a list of some key natural enemies:(a)Predators

Clerid Beetles (*Cleridae* family): Species such as *Thanasimus dubius* are known predators of bark beetles, including *Xyleborus*. They actively hunt adult beetles and larvae in their galleries [71,72].

Ants (*Formicidae* family): Plagiolepis sp. is an indigenous predator of the coffee twig borer, *Xylosandrus compactus*, and shows potential as a biological control agent for this pest in Ugandan coffee plants [73]. 

(b)Parasitoids

*Braconid* Wasps (*Braconidae* family): Parasitoid wasps like *Dendrosoter sulcatus* are known to parasitize the larvae of *Xyleborus* beetles. They lay eggs inside the beetle larvae, and the wasp larvae consume the beetle from within.

Eulophid Wasps (*Eulophidae* family): Some species within this family are known to parasitize ambrosia beetle larvae.

*Pteromalid* Wasps (*Pteromalidae* family): These wasps can parasitize various stages of *Xyleborus* beetles, primarily targeting the larvae within their galleries [74].

(c)Pathogens

Fungal Pathogens (e.g., *Beauveria bassiana*): This entomopathogenic fungus infects and kills *Xyleborus* beetles, reducing their populations. It can be naturally present in the environment or applied as a biocontrol agent [9].

(d)Nematodes: Certain species of nematodes, such as *Steinernema* and *Heterorhabditis*, can parasitize and kill *Xyleborus* beetles. They enter through natural openings and release bacteria that kill the beetles [75]. (e)Woodpeckers: These birds can excavate infested wood to reach and feed on both adult *Xyleborus* beetles and their larvae. Woodpeckers are natural predators of many bark beetle species, including ambrosia beetles. For instance, particularly Three-toed Woodpeckers, play a role in regulating bark beetle populations in coniferous forest landscapes [76].

## 3. Economic and Ecological Impact

The redbay ambrosia beetle, *X. glabratus*, can affect the economy and ecology due to their role as a vector of plant pathogenic fungi, leading to damage to various tree species and crops. Their role as decomposers contributes to nutrient cycling and forest health, but their ability to infest healthy trees and introduce pathogenic fungi can lead to economic losses and ecological disruptions. Effective management strategies are essential to mitigate these impacts and protect both economic resources and ecosystem health [36,77].

### 3.1. Economic Impact

*Xyleborus* beetles have an economic impact, particularly in regions where forestry and agriculture are major economic drivers [78]. In the forestry sector, these beetles cause losses estimated at USD 1000/acre in orchards infested wood often suffers from structural damage and staining due to the fungal symbionts introduced by the beetles. This staining, commonly referred to as “ambrosia stain,” can reduce the market value of timber, even if the structural integrity remains intact. Additionally, severe infestations can lead to tree mortality, necessitating costly management interventions such as tree removal and replacement [2,3,19].

In the agricultural sector, *Xyleborus* beetles pose a threat to fruit trees and other crops. For example, the redbay ambrosia beetle, *X. glabratus* is a pest in avocado orchards in the southeastern United States. This beetle transmits the fungus *R. lauricola*, which causes laurel wilt, a disease that has led to extensive tree mortality in commercial avocado plantations. The economic losses associated with laurel wilt include reduced yields, increased management costs, and the loss of marketable fruit, which can have far-reaching impacts on the profitability of avocado production [47,79].

The economic impact of *Xyleborus* beetles, particularly through laurel wilt disease affecting the avocado industry, is serious. Annual potential sales losses in southern Florida’s avocado industry could reach up to USD 30 million, with 75% and 50% crop reductions leading to USD 22.5 million and USD 15 million in losses, respectively. Property values of avocado groves could plummet by about USD 326 million if all bearing trees are destroyed. Additionally, increased management and control costs, including prophylactic treatments and intensified monitoring, could raise annual expenses by approximately USD 4.5 million. Overall, the Florida avocado industry faces an estimated annual economic impact of around USD 100 million due to this disease [78,80,81].

Other crops impacted by ambrosia beetles and their associated fungi include citrus, grapevine, cacao, coffee, macadamia, peach, and tea. These beetles, by shifting from dead or declining trees to healthy ones, pose a threat to these crops, leading to decreased yields and increased management costs [18].

For example, *X. ferrugineus* is a species capable of infesting wood pieces stored in fields and sawmills, as well as piles of freshly cut and moist wood. It causes the death of apparently healthy trees by introducing fungi that cause vascular wilts, such as *Ceratocystis fimbriata*, which can lead to the death of cacao trees [69,82]. Similarly, *X. affinis* infests tropical woods and exhibits similar behavioral characteristics and effects, impacting cacao and mango crops and trees within the laurel family [16,60].

The global trade in timber and ornamental plants has facilitated the spread of *Xyleborus* beetles to new regions, where they can become invasive pests. Invasive *Xyleborus* species, such as *X. glabratus* and *X. similis*, have established populations outside their native ranges, causing economic damage to local forestry and agriculture industries. The introduction of these beetles often leads to increased management costs, including monitoring, quarantine measures, and chemical treatments to control their spread [50].

### 3.2. Ecological Impact

Invasive species of *Xyleborus* can alter the ecosystems they invade by competing with native species, modifying community structures, and spreading dangerous pathogens since they carry and cultivate fungus as a resource of food for their progeny [3]. Therefore, it is crucial to understand and manage these impacts to protect the biodiversity and integrity of affected ecosystems.

Invasive species of the genus *Xyleborus* have demonstrated impacts on the ecosystems they colonize. Scientific studies have documented how these species disperse intercontinental, negatively affecting local communities. Laurel Wilt Disease has extirpated populations of Lauraceae species across the southeastern United States, including *Persea borbonia* L. Spreng., *Persea palustris* Raf., *Lindera benzoin* (L.) Blume, *Sassafras albidum* (Nuttall) Nees, *Persea americana* Mill, affecting coastal lowland ecosystems from North Carolina to Texas [36,83,84]. These invasions can alter the composition of native communities and compete with native species for essential resources, resulting in decreased biodiversity and changes in ecosystem structure [85].

Additionally, a study about the potential invasion of exotic ambrosia beetles, such as *Xyleborus glabratus*, in various ecosystems used ecological niche models to predict their impacts. This study indicates that these beetles can establish themselves in a variety of habitats, spreading pathogens like the fungus *R. lauricola*, which causes the disease known as “laurel wilt.” This disease severely affects trees in the Lauraceae family, posing an ecological risk to these ecosystems [51].

Moreover, the fungi introduced by *Xyleborus* beetles can also affect the microbial communities within the infested trees. The fungal symbionts can outcompete native fungi and other microorganisms, leading to shifts in the microbial diversity of the wood. These changes can affect the decomposition processes and nutrient cycling within the forest ecosystem, further impacting the overall health and biodiversity of the forest [2,86].

### 3.3. Ecosystem Services

The activities of *Xyleborus* beetles can have both positive and negative impacts on ecosystem services. On the positive side, their role in the decomposition of dead wood contributes to nutrient cycling and soil formation, essential processes for maintaining forest productivity and health. Their burrowing activities create microhabitats that support a diverse array of organisms, enhancing ecosystem resilience. These services include carbon sequestration, air and water purification, climate regulation, and the provision of habitat for wildlife [86,87].

Moreover, the loss of these services due to disruptions caused by invasive beetles can have implications for both the environment and human well-being. The decomposition process facilitated by these beetles helps in nutrient cycling and soil formation, maintaining the productivity and health of forests [10]. Additionally, the creation of microhabitats by their burrowing activities supports biodiversity and ecosystem resilience, contributing to essential services like carbon sequestration and climate regulation [11].

Understanding and managing the impact of *Xyleborus* beetles is crucial to preserving these vital ecosystem functions and mitigating the broader environmental and societal consequences of their activities.

## 4. Control and Management Strategies

### 4.1. Preventive Measures

Preventive measures methods focus on minimizing the conditions favorable for *Xyleborus glabratus* establishment and spread. Removing and properly disposing of infested plant material is crucial. Sanitation practices, such as chipping and burning infected wood, help reduce beetle populations and prevent disease spread. Additionally, maintaining tree health through proper irrigation and fertilization can make trees less susceptible to beetle attacks. Practices such as removing infested trees and destroying them promptly can significantly reduce the spread of laurel wilt [58,88].

Understanding the host selection behavior of *X. glabratus* is crucial for developing effective management strategies. Studies have shown that *X. glabratus* uses the diameter of the host tree’s stem as a visual cue to select its host. Beetles are more likely to infest trees with larger diameters, indicating that tree size is an important factor in host selection [53]. With the knowledge of these preferences, it is possible to develop management strategies that target high-risk trees. Effective management of *X. glabratus* requires diligent monitoring and early detection. Continuous surveillance programs are essential in areas with established populations and regions at risk of infestation. Trap-based monitoring systems using lure-baited traps can detect beetle presence before the onset of laurel wilt symptoms in host trees [89].

Various types of lures have been tested, including those based on manuka oil, phoebe oil, cubeb oil, and α-copaene, derived from different botanical sources. Manuka oil, extracted from the New Zealand tea tree (*Leptospermum scoparium*), and phoebe oil, derived from Brazilian walnut (*Phoebe porosa*), have been effective in attracting beetles. Cubeb oil, derived from the cubeb pepper plant (*Piper cubeba*), and α-copaene, a sesquiterpene found in various plants, are particularly potent attractants. Cubeb oil and α-copaene traps have demonstrated higher capture rates of *X. glabratus*, making them favorable for monitoring programs [90,91].

Additionally, studies have been conducted using volatile substances from different sources. Volatile compounds from the leaves of redbay (*Persea borbonia*) have been found to attract *X. glabratus*, highlighting the role of host plant volatiles in beetle behavior [92]. Research has also shown that volatile organic compounds produced by the laurel wilt pathogen, *Raffaelea lauricola*, can attract *X. glabratus*, suggesting a symbiotic relationship between the beetle and the fungus [53]. Regular monitoring using these lures allows for timely intervention and the implementation of control measures to prevent further spread.

Experiments comparing the attraction of *X. glabratus* to avocado wood and litchi wood demonstrated that the beetle is more attracted to litchi wood. This increased attraction is attributed to the higher content of α-copaene in litchi wood compared to avocado wood. This finding underscores the importance of α-copaene in the beetle’s host selection process and provides insights for developing more effective lures [92,93].

Studies have also explored the use of eucalyptol as a lure. Field trials demonstrated that traps baited with eucalyptol captured significant numbers of *X. glabratus*, underscoring the compound’s attractiveness to the beetle. This is particularly relevant for avocado trees, as variations in eucalyptol content among different avocado cultivars may influence their susceptibility to the beetle. Some varieties of avocado might have higher eucalyptol levels, making them more attractive to *X. glabratus* and thereby more susceptible to infestation and subsequent laurel wilt infection [39,91].

### 4.2. Chemical Control

Chemical control involves the use of insecticides and fungicides to protect high-value trees from *Xyleborus glabratus* and laurel wilt. Preventive treatments with systemic insecticides like bifenthrin and permethrin have shown effectiveness in reducing beetle populations and preventing new infestations [94]. Fungicide treatments, particularly with propiconazole, have demonstrated efficacy in protecting trees from laurel wilt when applied through root flare injections. However, these treatments require regular reapplication and may not be feasible on a large scale due to cost and potential environmental impacts [95].

Recent studies have been focused on evaluating possible repellents for *X. glabratus.* Cloonan et. al. (2023) compared the performance of Piperitone to that of the known repellent verbenone to assess its potential as a cost-effective alternative. Piperitone showed significant repellent effects against *X. glabratus*, comparable to verbenone, but at potentially lower costs. The study suggests that piperitone could be an effective tool for integrated pest management (IPM) programs aimed at controlling *X. glabratus* and managing laurel wilt [96]. 

Due to most of the beetle population being found within infected trees, insecticides have shown little effectiveness in controlling the pest. However, it is important to apply these insecticides around the trunks of infected trees to prevent their movement. Malathion, Danitol^®^ (fenpropathrin, Bayer, S.A., Santiago, Chile), and Epi-merk^®^ (abamectin, Merck & Co., Inc., Rahway, NJ, USA) are the most used insecticides. There are also adjuvants like NuFilm^®^ (Miller Chemical, Atlanta, GA, USA) that improve the efficacy of the insecticides [97]. Additionally, there are studies on insecticides that have been evaluated in combinations, such as zeta-cypermethrin + bifenthrin and lambda-cyhalothrin + thiamethoxam, which provides useful tools for integrated pest management strategies [94]. However, the variable effectiveness of other insecticides highlights the need for further research and the development of more robust control measures.

Our research group is also working on chemical control strategies to mitigate the risks posed by *Xyleborus* species, specifically through the development of chiral nitroguanidine-based compounds. These compounds have shown promise in targeting these beetles, aiming to enhance the efficacy of pest control methods while minimizing environmental impact [21,22,23,98,99].

### 4.3. Biological Control

Biological control methods are being explored to manage *Xyleborus glabratus* populations. Entomopathogenic fungi (EPF) such as *Beauveria bassiana* and *Isaria fumosorosea*, have shown promise in laboratory and field trials by causing significant mortality in beetle populations [58]. Research is ongoing to optimize delivery methods and assess the long-term effectiveness of these biological agents in various environmental conditions. Additionally, biological control agents, such as parasitoids and predators, are being investigated for their potential to reduce beetle populations naturally [96].

Research has shown that several strains of *Beauveria bassiana* infect more than 12 species of bark beetles, indicating its broad-spectrum potential. Moreover, field studies have demonstrated that *B. bassiana* and *Metarhizium brunneum* can significantly reduce beetle populations, suggesting that these fungi could be integrated into pest management programs for *X. glabratus* [88].

In addition to these studies, research on *Xyleborus affinis*, a closely related species, has provided valuable insights into the effectiveness of *Beauveria bassiana* as a biological control agent. This study highlighted that *B. bassiana* not only caused significant mortality in *X. affinis* but also exhibited potential for horizontal transmission among beetle populations. The findings suggest that similar strategies could be effective against *X. glabratus*, offering a broader application of EPF in managing ambrosia beetles [88].

A study by Peña et al. (2013) highlighted the role of predators and parasitoids associated with Scolytinae in Persea species in both Florida and Taiwan. Predators identified included species from the families *Laemophloeidae*, *Staphylinidae*, *Zopheridae*, and *Monotomidae*, while parasitoids from the families *Braconidae*, *Eulophidae*, *Pteromalidae*, *Encyrtidae*, *Eupelmidae*, and *Bethylidae* were also found. Notably, in the context of *X. glabratus*, *Bethylidae*, *Braconidae*, *Encyrtidae* (possibly *Closterocerus* sp.), and *Scelionidae* were observed emerging from infested wood [100].

The development of host resistance through breeding programs is a critical long-term strategy. Efforts are underway to identify and propagate tree genotypes that exhibit resistance to laurel wilt. Preliminary studies have identified red bay and swamp bay clones with tolerance to the disease, and these clones are being evaluated in field trials [29]. Conservation and propagation of resistant germplasm are essential components of integrated management strategies. Resistant varieties of avocado and other lauraceous hosts are being developed to provide long-term solutions for managing laurel wilt [101].

### 4.4. Integrated Pest Management (IPM)

An Integrated Pest Management (IPM) approach combines multiple control methods to manage *X. glabratus* and laurel wilt effectively. IPM strategies include monitoring and scouting, sanitation, chemical, and biological control, and the development of resistant host plants. Public education and outreach are also vital to prevent the spread of the beetle and disease through human activities, such as the movement of infested firewood. IPM emphasizes the integration of these methods to achieve sustainable and effective management while minimizing environmental impact [102].

These beneficial effects of applying IPM strategies have been observed in other pests caused by ambrosia beetles. *Trypodendron lineatum* (striped ambrosia beetle), is a significant pest in British Columbia’s forest industry. The IPM program for ambrosia beetles in British Columbia began in 1981, with the cornerstone being the use of semiochemical-based mass trapping, particularly for *T. lineatum* [103]. This program was developed as an alternative to the previous reliance on chemical insecticides, such as DDT and lindane, which were phased out due to environmental concerns. Despite the initial costs of setting up and maintaining these traps, the long-term savings and reduction in pest damage have proven the IPM strategy to be cost-effective [104]. Additionally, the environmental benefits of reducing chemical pesticide use have added value to this approach.

In Sri Lanka, tea plants (*Camellia sinensis*) were affected by *Xyleborus fornicatus*, and various IPM techniques such as biological control, cultural practices, mechanical control, and the judicious use of chemical pesticides were employed for the management of *X. fornicatus* [105]. Selection and planting of tea cultivars that are either resistant or tolerant to *X. fornicatus* is a primary strategy for control of *X. fornicatus*, also, proper pruning techniques and sanitation measures are critical in reducing the wood rot that can occur due to fractures and wounds induced by *X. fornicatus*. This includes removing affected wood and applying wound dressings to large prune cuts to prevent further damage [106].

## 5. Research and Advances

Recent years have witnessed advancements in the study of *Xyleborus* beetles, with an increasing number of research articles being published annually. This trend reflects the growing recognition of the ecological and economic impact of these beetles and the urgent need for effective management strategies. Figure 2 illustrates the surge in research activities over different periods, highlighting the accelerated pace of scientific inquiry into the biology, behavior, and control of *Xyleborus* beetles. The exponential growth in publications underscores the importance of continuous research to address the challenges posed by these pests and to develop sustainable solutions for their management.

### 5.1. Molecular and Genetic Studies

The genetic variability of *X. glabratus* populations across Asia was assessed using COI (mitochondrial DNA) and CAD (nuclear DNA) markers, revealing diverse genetic types and leading to the identification of two new species, *X. insidiosus* and *X. mysticulus*, based on phylogenetic and morphological data. Additionally, sexual dimorphism in mandible size and shape in *X. affinis* was linked to temperature-induced phenotypic variation, potentially impacting ecological activities like feeding. For *X. principalis*, gene sequence analysis showed significant variation in the COI gene (up to 14.2% divergence), but minimal variation in the 28S gene, with no geographic or consistent morphological patterns, indicating *X. principalis* is monophyletic but lacks consistent differentiation [5,6].

### 5.2. Behavioral Studies

*X. affinis* is one of the ambrosia beetle species potentially considered harmful. However, observations indicate that tunnel excavation, once the species is established in the trunk, is an exclusive task of the females, who extend the gallery where they live. The larvae, females, and adult males graze and feed on fungi that cover the gallery walls. In a blocking function, the females remain inactive to use their bodies to protect the tunnels. Activities related to nest care include grooming the nest and individuals, cultivating fungi, and caring for the brood. Cannibalism is an observed behavior where they feed on larvae, pupae, or dead adults to maintain hygiene in the nest [33].

In the galleries of *X. bispinatus*, eggs are laid, and the larvae feed on fungi cultivated by the females on the gallery walls. These same females are responsible for constructing the main and secondary tunnels. The length of these tunnels is related to the number of adults, suggesting that gallery extension is vital for available food. Laying eggs at the farthest ends of the galleries may represent an adaptation for optimal use of space and food resources. The founding females seem to keep the main gallery clear to allow it to be used as a corridor to manage ambrosia gardens, and care for the eggs, and larvae. In their symbiotic relationship with fungi, *Raffaelea arxii* and *Raffaelea subfusca* are the most prevalent and are cultivated in the galleries, as they are essential for the nutrition of larvae and adults [107].

Ambrosia beetles, such as *X. affinis*, engage in symbiotic mutualisms with fungi for nutritional purposes. The practice of fungal farming involves selecting beneficial fungi as the main food source, protecting these fungi from pathogens, and providing the necessary nutrients for their growth. Bacteria play an important role in the agricultural practices of ambrosia beetles, helping to defend fungal crops, fix atmospheric nitrogen, degrade plant biomass, and plant defenses. The metabolic capabilities of bacteria and yeasts are crucial for supporting the beetle-fungus agricultural symbiosis, especially in the early stages of gallery development. Bacteria and fungi provide essential nutrients that the beetles need for their development, influencing their behavior and reproductive success [44].

## 6. Future Directions

*Xyleborus* beetles, known for their symbiotic relationship with fungi, have a unique life cycle that allows them to thrive across diverse environments. This mutualism is integral to their survival, enabling rapid population growth and colonization of new areas. Their reproductive strategies and habitat preferences have been well-documented, with a particular focus on the impact of invasive species such as *X. glabratus* and *X. similis*. Recent research has also highlighted the economic and ecological damage caused by these beetles, particularly through the introduction of pathogenic fungi that disrupt forest ecosystems and agricultural productivity [78,80,97].

Despite advancements in understanding the biology and ecology of *Xyleborus* beetles, several critical gaps remain. The intricate relationship between beetles and their fungal symbionts poses challenges for developing targeted management strategies that do not adversely affect other ecosystem components. Furthermore, the rapid adaptation of invasive species in non-native regions underscores the need for more stringent quarantine and monitoring measures. Although there have been promising developments in chemical and biological control methods, there is still a need for optimized and sustainable Integrated Pest Management (IPM) strategies that can be adapted to different environmental conditions [42,98]. The spread of diseases like laurel wilt underscores the broader environmental implications of *Xyleborus* infestations. Effective management of these beetles requires an integrated approach that combines preventive measures, chemical control, biological control, and IPM strategies. Further exploration into the genetic diversity and population structure of *Xyleborus* beetles is crucial. Advanced molecular tools, such as DNA barcoding and genome sequencing, can deepen our understanding of beetle-fungal interactions and support the development of targeted biocontrol agents [5,6,29,108].

Investigating the behavioral ecology of *Xyleborus* beetles, particularly their host selection mechanisms and mating behaviors, will be essential for improving detection and control methods. Research on volatile organic compounds that influence beetle attraction and repulsion can lead to the development of more effective lures and repellents. There is a need for continued research into the integration of various control methods, including chemical, biological, and cultural approaches, into cohesive IPM strategies. Evaluating the effectiveness of these combined measures in different environmental contexts will help develop robust and adaptable IPM programs. Conservation of resistant tree genotypes and the development of breeding programs for disease-resistant varieties are critical for long-term management. Identifying and propagating resistant germplasm will provide sustainable solutions to mitigate the impact of *Xyleborus* beetles on forestry and agriculture. The ongoing threat posed by *Xyleborus* beetles necessitates continuous research and collaboration among scientists, policymakers, and industry stakeholders. By addressing the identified knowledge gaps and pursuing targeted research in the areas outlined above, it will be possible to develop more effective and sustainable management strategies. These efforts will be essential in mitigating the economic and ecological damage caused by these beetles while preserving forest ecosystems and ensuring agricultural productivity.

## Figures and Tables

**Figure 1 insects-15-00706-f001:**
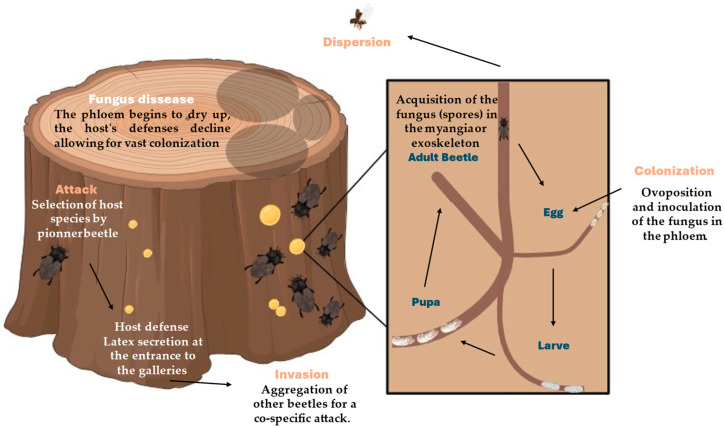
Life cycle and reproduction of *Xyleborus*.

**Figure 2 insects-15-00706-f002:**
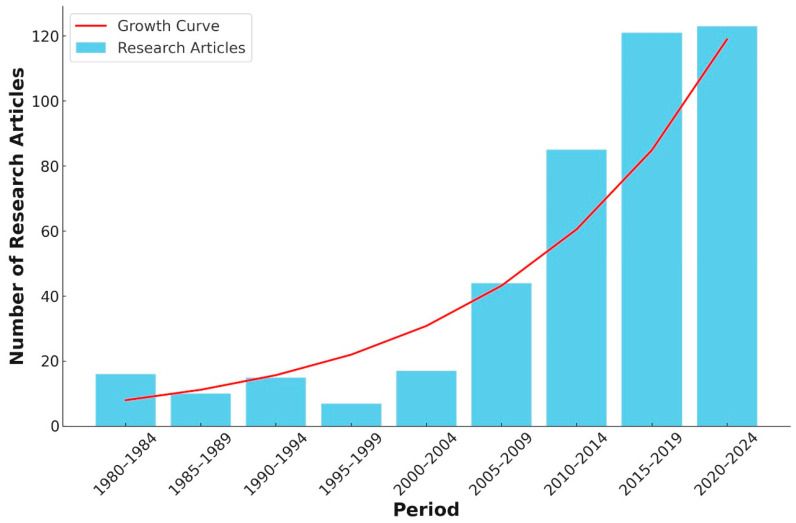
Number of research articles published on *Xyleborus* beetles from 1980 to 2024, aggregated in 5-year periods. Data for this graph were collected from SciFinder, Web of Science, and Scopus.

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
