# Peer review of "The Biology, Impact, and Management of Xyleborus Beetles: A Comprehensive Review"

_insects, 2024, doi:10.3390/insects15090706_

Round 1
Reviewer 1 Report
Comments and Suggestions for Authors
In the paper "The Biology, Impact, and Management of Xyleborus Beetles: A Comprehensive Review," the authors Rodriguez-Becerra et al. address a very interesting issue related to ambrosia beetles and explain why such a review is necessary. Generally, the article is well-written; however, there is a lot of repetition throughout various sections, raising the question of whether the content could have been simplified and written more concisely. On the other hand, important sections are missing, such as the position of this group of insects among bark beetles, etc.
Therefore, I suggest that the authors make significant revisions:
- In the Introduction, the position of Xyleborus bark beetles should be presented by listing the different groups of bark beetles and indicating where Xyleborini are situated, as well as which other genera belong to the Ambrosia group.
- In the section "Biology and Ecology of Xyleborus," natural enemies are missing. This is a review article, and I believe this topic cannot be overlooked, especially since biological control and some predator and parasitoid species are later mentioned. This is an opportunity to list the natural enemies systematically.
- In section 2.1.2, "Fungal Inoculation," the location of mycangia is mentioned very briefly, only stating that it depends on the species. Here, more examples should be provided, and the discussion on mycangia should be more comprehensive.
- Figure 2 seems to delve unnecessarily into detail, and I would recommend omitting it.
- Sections 6. Discussion and 7. Conclusion mostly repeat information that has already been presented. I recommend that the authors consider removing these sections and instead title the subsection "Future Prospects." In this case, the structure should address the current state of knowledge, knowledge gaps, and what specifically needs further investigation. These points are already mentioned in the text, but a different structure is needed.
Author Response
Thank you for your valuable feedback on my manuscript, titled "[ The Biology, Impact, and Management of Xyleborus Beetles: A Comprehensive Review t]." I have carefully considered and addressed all the comments and suggestions provided. I hope that the revisions made enhance the quality of the manuscript, making it suitable for publication.
Please find the revised version attached for your review. I appreciate your time and effort in evaluating my work.
Thank you for your consideration.
Here are our actions to your comments/suggestion:
GENERAL COMMENTS
In the paper "The Biology, Impact, and Management of Xyleborus Beetles: A Comprehensive Review," the authors Rodriguez-Becerra et al. address a very interesting issue related to ambrosia beetles and explain why such a review is necessary. Generally, the article is well-written; however, there is a lot of repetition throughout various sections, raising the question of whether the content could have been simplified and written more concisely. On the other hand, important sections are missing, such as the position of this group of insects among bark beetles, etc.
Therefore, I suggest that the authors make significant revisions:
Comment 1
- In the Introduction, the position of Xyleborus bark beetles should be presented by listing the different groups of bark beetles and indicating where Xyleborini are situated, as well as which other genera belong to the Ambrosia group.
Response 1
Thank you for your valuable feedback. In response to your suggestion, I have added a figure that outlines the scientific classification of the bark beetles and clarifies the position of Xyleborini within this classification.
Comment 2
- In the section "Biology and Ecology of Xyleborus," natural enemies are missing. This is a review article, and I believe this topic cannot be overlooked, especially since biological control and some predator and parasitoid species are later mentioned. This is an opportunity to list the natural enemies systematically.
Response 2
We have added section 2.3 which includes some of the natural enemies of the Xyleborus beetles
Comment 3
- In section 2.1.3, "Fungal Inoculation," the location of mycangia is mentioned very briefly, only stating that it depends on the species. Here, more examples should be provided, and the discussion on mycangia should be more comprehensive.
Response 3
We have added a paragraph with some examples about the mycangia and added how environmental factors can affect fungal diversity.
Comment 4
- Figure 2 seems to delve unnecessarily into detail, and I would recommend omitting it.
Response 4
We have removed Fig 2
Comment 5
- Sections 6. Discussion and 7. Conclusion mostly repeat information that has already been presented. I recommend that the authors consider removing these sections and instead title the subsection "Future Prospects." In this case, the structure should address the current state of knowledge, knowledge gaps, and what specifically needs further investigation. These points are already mentioned in the text, but a different structure is needed.
Response 5
We have renamed sections 6 and 7 as recommended by the reviewer. Now, future directions address the current state-of-the-art of Xyleborus beetles.
Reviewer 2 Report
Comments and Suggestions for Authors
This is a much needed review about the life cycle, mitigation and economic importance of Xyleborus beetles. Generally, the text is well prepared. I have some critical remarks, such that - if successfully overcome - would hopefully make the manuscript considerably stronger.
1. How was literature collected? What were the used search engines, search terms, and dates? Seems as if this is rather a narrative, non-systematic paper, though Fig. 3 suggests that the searches might have been done in a systematic way. Please add these (and perhaps Fig. 3 too) as chapter 2.
2. The text contains lots of unwarranted words, such that are entirely subjective; the whole text should be carefully checked and all such words removed. A few examples (Abstract) significant, considerable, valuable; (Introduction) fascinating, significant, important, substantial, profound, paramount; (3.1 Economic impact) major threat, significant, substantial damage, etc. are just empty buzzwords if nothing concrete (hard facts, numbers, shares...) is given. Sometimes these might be warranted, but even then you should provide the reader quantitative numbers.
3. Terminology may warrant fine tuning. Beetle impacts on trees are "damage" only from an economic perspective - they are crucial parts of healthy ecosystem functioning. (With "healthy" I refer to ecosystems where human impact is minimal.) Also, beetles are "pests" or have "destructive potential" mostly only from an economic viewpoint - just talk about "insects" or "bark beetles" as appropriate. In biological control, or from forestry perspective, beetle impact might be referred to as "mass occurrence of beetles".
4. The life history section (starting from 2.1) would greatly benefit from adding research findings about duration of each life phase, their timing (months), one-/multivoltine development, temperature tolerance, possible cases of adaptation to new host tree species. Also, possible geographic variation in host plants and fungal associates would be of interest for foresters and ecologists. You have nicely compiled Table 1 that shows host tree species, but I suggest compile another table showing these "hard numeric facts" about life history of the assessed species, as much as possible based on literature.
5. Chapter 2.2 is difficult to grasp. The text is not very precise, i.e., it does not provide hard facts (see comment #4). For instance, in its opening paragraph, what does "various environments" mean, and what do you mean with environment (how many of such can different species use, based on research)? Section 2.2 does not say anything about habitats and sistributions of these beetles.
6. Table 1 rises a question that might warrant discussion, at least if literature exists: some species have been found in two or more continents; are they really the same species, or is there evidence for (human) introductions and, if so, are mass occurrences more common or more "damaging" in the new region?
7. Citing literature seems quite vague and should be strengthened. Some examples are shown below. The text contains lots of sentences without clarity on where the information comes.
8. Other, perhaps more minor details
Opening paragraph of 2.1 does not contain references. Please add.
Fig. 1 may not be readable if printed black-and-white. Consider lightening the colors and enlarging the fonts. Also, please check the figure for typos (there are several).
Chap 2.1.7, how common is the co-occurrence of different developmental phases? Also, the opening paragraph of this section does not contain references, please add.
Table 1 should be referred to in several text occasions, whenever you write about fungal associates, host trees or geography. Currently it seems it has not been cited anywhere (though I may have missed such).
Chap 2.2.5 does not say anything about conservation. Please add text and references about (possible) impacts of Xyleborus (or other bark beetle?) impacts on conservation-relevant environmental conditions or other species groups in the mass-occurrence cites, if any.
Chap 2.3 introduction part repeats what you have said earlier on, please delete.
Chapters 2.3.1, 2.3.2, 2.3.3 and 2.3.4 repeat earlier text. Delete or combine with earlier text.
Chap 2.3.5 introductive paragraph also repeats earlier text, delete. However, the next paragraph appears useful.
Chap 3 does not say much (if anything) about ecological impact. If this viewpoint is to be kept, it must be expanded considerably, to make it equal to the economic viewpoint (or as much as bark beetle literature allows). Discussion (Chap 6) contains some notes and references which could/should be presented here.
Chap 3.1 would benefit from adding concrete values to strengthen the economic importance. Paragraph 3 shows some for agriculture, but there is nothing about forestry.
Chap 3.2: (i) you mention dangerous pathogens - dangerous for what and how, any evidence showing the danger? (ii) next paragraph says "scientific studies" but only one study is cited. (iii) what are "local communities" (human populations, species assemblages...)?
Chap 4.2 would be strengthened by providing numerical facts about beetle death rates if the mentioned chemicals are used. Also, do you have information about the efficiency of biological control?
Subtitle 4.3 is merged with Fig. 2 text.
Page 12, only species and genus names are italicized.
Chap 5 (particularly Fig. 3!) might better fit as Chapter 2 (presentation of literature collecting). And/or actually much of these things are described elsewhere earlier on and could/should be combined. No need to distinguish "old" and "recent" research - the reader can judge that by looking at references.
Discussion would be stronger if it more tihhtly focused on your review findings. The meta level visions about the importance of DNA etc. should be kept to a minimum. You could pinpoint research knowledge gaps from results described above.
Conclusions should be clarified to closely match review main results.
List of references seems good, but it is difficult to really assess it because the journal uses numbers instead of alphabetical order for references. This is of course the journal's fault, not authors'.
Comments on the Quality of English LanguageNothing in particular. The text is mostly clear but it has currently other issues as described above.
Author Response
Dear Reviewer,
Thank you for your valuable feedback on my manuscript, titled, The Biology, Impact, and Management of Xyleborus Beetles: A Comprehensive Review. I have carefully considered and addressed all the comments and suggestions provided. I hope that the revisions made enhance the quality of the manuscript, making it suitable for publication.
Please find the revised version attached for your review. I appreciate your time and effort in evaluating my work.
Thank you for your consideration.
GENERAL COMMENTS
This is a much needed review about the life cycle, mitigation and economic importance of Xyleborus beetles. Generally, the text is well prepared. I have some critical remarks, such that - if successfully overcome - would hopefully make the manuscript considerably stronger.
Comment 1
- How was literature collected? What were the used search engines, search terms, and dates? Seems as if this is rather a narrative, non-systematic paper, though Fig. 3 suggests that the searches might have been done systematically. Please add these (and perhaps Fig. 3 too) as chapter 2.
Response 1
The revision used the SciFinder, Scopus, and WebofScience databases to search and analyze the state-of-the-art Xyleborus beetles. The main search engines were, Xyleborus, Bark beetles, Ambrosia beetles, etc. Regarding the search terms, we used Xyleborus classification, management, evolution, and diversity, among others. All the searches were from 1980 to date.
Comment 2
- The text contains lots of unwarranted words, such that are entirely subjective; the whole text should be carefully checked and all such words removed. A few examples (Abstract) significant, considerable, valuable; (Introduction) fascinating, significant, important, substantial, profound, paramount; (3.1 Economic impact) major threat, significant, substantial damage, etc. are just empty buzzwords if nothing concrete (hard facts, numbers, shares...) is given. Sometimes these might be warranted, but even then you should provide the reader quantitative numbers.
Response 2
We have revised the whole document and removed all the subjective words.
Comment 3
- Terminology may warrant fine tuning. Beetle impacts on trees are "damage" only from an economic perspective - they are crucial parts of healthy ecosystem functioning. (With "healthy" I refer to ecosystems where human impact is minimal.) Also, beetles are "pests" or have "destructive potential" mostly only from an economic viewpoint - just talk about "insects" or "bark beetles" as appropriate. In biological control, or from forestry perspective, beetle impact might be referred to as "mass occurrence of beetles".
Response 3
We agree with the reviewer's comment. Insects play an important role in the ecosystem and not all of them can be classified as pests. We have edited our manuscript to clearly state that the evidence suggests that the Xyleborus beetle can cause economic and ecologic damage to forests and important agricultural orchards.
Comment 4
- The life history section (starting from 2.1) would greatly benefit from adding research findings about duration of each life phase, their timing (months), one-/multivoltine development, temperature tolerance, possible cases of adaptation to new host tree species. Also, possible geographic variation in host plants and fungal associates would be of interest for foresters and ecologists. You have nicely compiled Table 1 that shows host tree species, but I suggest compile another table showing these "hard numeric facts" about life history of the assessed species, as much as possible based on literature.
Response 4
We agree with the reviewer´s comments. We have added extra references to strengthen this section. It is important to mention that most of the comments of the reviewer have been shown by Lira-Noriega, Scientific Reports, (2018) 8:10179. DOI:10.1038/s41598-018-28517-4
Comment 5
- Chapter 2.2 is difficult to grasp. The text is not very precise, i.e., it does not provide hard facts (see comment #4). For instance, in its opening paragraph, what does "various environments" mean, and what do you mean with environment (how many of such can different species use, based on research)? Section 2.2 does not say anything about habitats and sistributions of these beetles.
Response 5
We thank the reviewer's comment, nonetheless, we do not agree. Section 2.2 refer to Table 1. This table shows not only the Xyleborus species but also the fungus-associated, affected plants, and distribution. Additionally, in the text, we wrote that the Xyleborus beetles are distributed globally, with species occurring in a wide range of climatic zones, from tropical rainforests to temperate woodlands.
Comment 6
- Table 1 rises a question that might warrant discussion, at least if literature exists: some species have been found in two or more continents; are they really the same species, or is there evidence for (human) introductions and, if so, are mass occurrences more common or more "damaging" in the new region?
Response 6
Indeed, this is an important observation. As we mentioned in the manuscript these invasions are often facilitated by human activities, such as the global trade of timber and ornamental plants, which can inadvertently transport beetles and their fungal symbionts to new areas, please see reference [40]
Comment 7
- Citing literature seems quite vague and should be strengthened. Some examples are shown below. The text contains lots of sentences without clarity on where the information comes.
Response 7
We have revised the manuscript to strengthen the text with literature cites.
Comments 8 and responses
- Other, perhaps more minor details
Opening paragraph of 2.1 does not contain references. Please add.
R = We have added the reference
Fig. 1 may not be readable if printed black-and-white. Consider lightening the colors and enlarging the fonts. Also, please check the figure for typos (there are several).
R = We have printed the image in black and white to ensure it can be seen and read
Chap 2.1.7, how common is the co-occurrence of different developmental phases? Also, the opening paragraph of this section does not contain references, please add.
R = We have added the reference and revised the paragraph
Table 1 should be referred to in several text occasions, whenever you write about fungal associates, host trees or geography. Currently it seems it has not been cited anywhere (though I may have missed such).
R = We have added the cites inside the table in brackets
Chap 2.2.5 does not say anything about conservation. Please add text and references about (possible) impacts of Xyleborus (or other bark beetle?) impacts on conservation-relevant environmental conditions or other species groups in the mass-occurrence cites, if any.
R = We have removed this section
Chap 2.3 introduction part repeats what you have said earlier on, please delete.
Chapters 2.3.1, 2.3.2, 2.3.3 and 2.3.4 repeat earlier text. Delete or combine with earlier text.
R = We have edited it to avoid repeated text
Chap 2.3.5 introductive paragraph also repeats earlier text, delete. However, the next paragraph appears useful.
R = We have deleted this section
Chap 3 does not say much (if anything) about ecological impact. If this viewpoint is to be kept, it must be expanded considerably, to make it equal to the economic viewpoint (or as much as bark beetle literature allows). Discussion (Chap 6) contains some notes and references which could/should be presented here.
R = We have added information on how this invasive species affects Lauraceaes mainly in the USA.
Chap 3.1 would benefit from adding concrete values to strengthen the economic importance. Paragraph 3 shows some for agriculture, but there is nothing about forestry.
R = The concrete values can be found in the text, we wrote:
The economic impact of Xyleborus beetles, particularly through laurel wilt disease affecting the avocado industry, is serious. Annual potential sales losses in southern Florida's avocado industry could reach up to $30 million, with 75% and 50% crop reductions leading to $22.5 million and $15 million in losses, respectively. Property values of avocado groves could plummet by about $326 million if all bearing trees are destroyed. Additionally, increased management and control costs, including prophylactic treatments and intensified monitoring, could raise annual expenses by approximately $4.5 million. Overall, the Florida avocado industry faces an estimated annual economic impact of around $100 million due to this disease [62,63].
Regarding forestry, we noted that the studies about economic impact are focused on agricultural activities rather than forestry.
Chap 3.2: (i) you mention dangerous pathogens - dangerous for what and how, any evidence showing the danger? (ii) next paragraph says "scientific studies" but only one study is cited. (iii) what are "local communities" (human populations, species assemblages...)?
R = We have added a text to indicate that the pathogen is the fungus that the beetle carries in their mycangia. Also, we have added more references. By local communities, we refer to the ecosystem that the beetle colonizers.
Chap 4.2 would be strengthened by providing numerical facts about beetle death rates if the mentioned chemicals are used. Also, do you have information about the efficiency of biological control?
Subtitle 4.3 is merged with Fig. 2 text.
R = The insecticides have demonstrated limited effectiveness in controlling the beetle. Peña's 2011 study suggests that reducing the number of entrance holes in trees may help decrease the severity of the infection. This highlights the need for developing new strategies to effectively control Xyleborus.
Regarding biological control, this is ongoing research reported in 2023, there is needed more studies to confirm that this strategy will be useful in the field.
Page 12, only species and genus names are italicized.
R = We have revised and corrected the text.
Chap 5 (particularly Fig. 3!) might better fit as Chapter 2 (presentation of literature collecting). And/or actually much of these things are described elsewhere earlier on and could/should be combined. No need to distinguish "old" and "recent" research - the reader can judge that by looking at references.
R = We thank the reviewer’s comments, however, we believe that the graph fits in this section since the objective is to show the evolution of the research and publications related to the topic.
Discussion would be stronger if it more tihhtly focused on your review findings. The meta level visions about the importance of DNA etc. should be kept to a minimum. You could pinpoint research knowledge gaps from results described above.
R = We agree with the reviewer’s comments. We have edited section 5.1
Conclusions should be clarified to closely match review main results.
R = We have added the section “future directions” to improve readability and understanding.
List of references seems good, but it is difficult to really assess it because the journal uses numbers instead of alphabetical order for references. This is of course the journal's fault, not authors'.
R = We are using the format of the journal

Round 2
Reviewer 1 Report
Comments and Suggestions for Authors
The article now looks better, and new sections have been added and updated. Table 1 was added based on the suggestion to better describe the position of Xyleborini; however, presenting it in a tabular format is unusual, and I wouldn't recommend this approach. The table can be described in words without going into too much detail (like in Table 1). I also notice that sections with scientific names have been added, but they are not written in italics
Author Response
Comment
The article now looks better, and new sections have been added and updated. Table 1 was added based on the suggestion to better describe the position of Xyleborini; however, presenting it in a tabular format is unusual, and I wouldn't recommend this approach. The table can be described in words without going into too much detail (like in Table 1). I also notice that sections with scientific names have been added, but they are not written in italics.
Responce
We agree with this comment. As requested by the reviewer, we have deleted Table 1 to include this information in the text. Besides, all the scientific names have been revised and written in italics.